# Novel Dual PI3K/mTOR Inhibitor, Apitolisib (GDC-0980), Inhibits Growth and Induces Apoptosis in Human Glioblastoma Cells

**DOI:** 10.3390/ijms222111511

**Published:** 2021-10-26

**Authors:** Wioleta Justyna Omeljaniuk, Rafał Krętowski, Wioletta Ratajczak-Wrona, Ewa Jabłońska, Marzanna Cechowska-Pasko

**Affiliations:** 1Department of Pharmaceutical Biochemistry, Medical University of Bialystok, 15-222 Bialystok, Poland; wioleta.omeljaniuk@umb.edu.pl (W.J.O.); r.kretowski@umb.edu.pl (R.K.); 2Department of Immunology, Medical University of Bialystok, 15-269 Bialystok, Poland; wioletta.ratajczak-wrona@umb.edu.pl (W.R.-W.); ewa.jablonska@umb.edu.pl (E.J.)

**Keywords:** apoptosis, glioblastoma, dual PI3K/mTOR inhibitor, GDC-0980, apitolisib

## Abstract

Deregulated PI3K/AKT/mTOR signalling commonly exists in glioblastoma, making this axis an attractive target for therapeutic manipulation. Given that activation of PI3K/AKT/mTOR promotes tumour growth, metastasis, and resistance to anticancer therapies, mTOR inhibitors show promise in the treatment of cancer. The aim of this study was to investigate the underlying mechanism of novel dual PI3K/mTOR inhibitor, Apitolisib (GDC-0980), in A-172 and U-118-MG GBM tumour cell line suppression. It has been demonstrated that GDC-0980 induces time- and dose-dependent cytotoxicity and apoptosis in investigated glioma cell lines. In our study, the strongest induction of apoptosis was exhibited in the A-172 line after 48 h of incubation with 20 µM GDC-0980, where we observed 46.47% of apoptotic cells. In conclusion, we first discovered that dual PI3K/mTOR blockade by GDC-0980 markedly suppressed survival of human GBM cells and induced apoptosis, independent of the ER stress-mediated DR5 activation. We suggest that GDC-0980, by exerting an inhibitory effect on PERK expression, may thus block its inhibitory effect on protein synthesis, leading to intensification of translation, and this may result in an increase in apoptosis. On the other hand, CHOP stimulates protein synthesis and increases apoptosis. These findings suggest that GDC-0980 may be a candidate for further evaluation as a chemotherapeutic agent for anti-GBM therapy.

## 1. Introduction

Glioblastoma multiforme (GBM) is the most aggressive and malignant tumour in the brain, characterised by infiltrating growth, intense migration, and rapid spread within the surrounding nervous tissue. Despite the standard treatment, including maximal surgical resection followed by adjuvant radiotherapy and chemotherapy, the prognosis of GBM patients remains poor [1,2]. The average life span, counted from diagnosis, is one year. Therefore, it is extremely important both to improve the therapy and search for the new therapeutic strategies [3,4]. 

Amplification of the gene encoding the epidermal growth factor receptor (EGFR) is commonly observed in GBM, leading to the activation of further kinases, including phosphatidylinositol 3′-kinase (PI3K), protein kinase B (AKT), and the mammalian target of rapamycin (mTOR). Serine-threonine kinase mTOR controls cell growth by regulating mRNA translation, metabolism, and autophagy [5]. The PI3K-AKTmTOR axis is the most frequently deregulated signalling pathway in human cancers [6,7]. 

PI3Ks are lipid kinases that generate phosphatidylinositol-3,4,5-tris phosphate (PIP3). PIP3 activates AKT, a serine/threonine kinase that inhibits apoptosis, promotes growth and drives proliferation. PIP3 also indirectly activates mTOR, a protein kinase critical for protein synthesis and cell growth (Figure 1) [5].

Deregulated PI3K/AKT/mTOR signalling commonly exists in glioblastoma, making this axis an attractive target for therapeutic manipulation. Given that activation of PI3K/AKT/mTOR promotes tumour growth, metastasis, and resistance to anticancer therapies, mTOR inhibitors show promise in the treatment of cancer. While inhibition of mTORC1 and mTORC2 may downregulate phosphorylation of AKT at Ser473 site, mTOR inhibitors can paradoxically enhance the PI3K/PDK1 axis. Thus, an inhibitor that targets both PI3K and mTOR may have superior anti-tumour activity compared to targeted mTOR alone [8,9]. Due to the similarity between PI3K and mTOR, some chemicals can inhibit both PI3K kinase and mTOR kinase [6]. Preclinical evaluation of dual PI3K/mTOR inhibitors, such as PI-103 and NVP-BEZ235, have demonstrated efficacy for these agents in blocking the growth of GBM cells in vitro and in vivo [10,11].

Apitolisib (GDC-0980) is dual PI3K/mTOR inhibitor, which can induce apoptosis of cancer cells and inhibit cancer cell growth (Figure 2) [12]. It is an orally available agent that targets PI3K and mTOR in the PI3K/mTOR signalling pathway [7]. It has been reported that GDC-0980 exhibits a broad spectrum of antitumour activity in many cancers and has been used in studies (Phase I/II) in the treatment of solid cancers. The strongest effect was observed in breast, prostate, and lung cancer cell lines, and this effect was less pronounced in pancreatic cancer and melanoma cell lines [1,7,13]. GDC-0980 has also been shown to inhibit tumour growth in xenograft models, including PI3K activated or loss of PTEN [7]. However, no clinical study on this inhibitor has been carried out in patients with GBM [1].

Inhibition of the activity of kinases belonging to the PI3K/AKT/mTOR pathway is an important goal in creating new options for the treatment of gliomas. In the present study, we aimed to elucidate the effects of the dual inhibitor Apitolisib (GDC-0980) on two established glioblastoma cell lines: A-172 and U-118-MG. Therefore, we blocked PI3K-mediated signalling with GDC-0980 at two distinct key points: the phosphorylation of the AKT signalling cascade and prevented its activation by reducing mTOR phosphorylation and investigated what effects this inhibitor had on the viability, proliferation, and apoptosis of investigated cells. 

## 2. Results

### 2.1. GDC-0980 Exerts Cytotoxic Effect on GBM Cells

First, we investigated the effective cytotoxic dose of GDC-0980 on the two different human GBM cell lines: A-172 and U-118-MG. The cells were treated with GDC-0980 at various concentrations ranging from 50 to 50,000 nM for 24 and 48 h and next subjected to MTT analysis (Figure 3A,B). The MTT test indicated that GDC-0980 reduced both A-172 and U-118-MG cell viability in a concentration-dependent manner after 24 and 48 h (Figure 3A,B). 

The proliferation assay by crystal violet staining was performed to further confirm the effect of GDC-0980 on GBM cell survival. We observed that the proliferation of A-172 (Figure 3C) and U-118-MG (Figure 3D) cells was inhibited by 10 and 20 µM GDC-0980 treatment (Figure 3C,D).

### 2.2. GDC-0980 Induces Apoptosis in GBM Cells

Flow cytometry was used to assess the apoptosis of A-172 and U-118-MG human GBM cell lines after 24 and 48 h of treatment with GDC-0980 (Figure 4). Live cells do not bind to Annexin V and PI (AnnexinV^−^/PI^−^), early apoptotic cells bind to Annexin V but not PI (AnnexinV^+^/PI^−^), late apoptotic cells bind to both molecules (AnnexinV^+^/PI^+^), and necrotic cells only bind to PI (AnnexinV^−^/PI^+^).

In our study, the percentage of apoptotic A-172 and U-118-MG cells was higher after GDC-0980 treatment in comparison to the control groups in a concentration- and time-dependent manner after 24 and 48 h of incubation (Figure 4A–D). Rates of apoptosis increased after using higher concentrations of GDC-0980 and longer incubation times compared to the corresponding control. The strongest induction of apoptosis was shown in line A-172 after 48 h of incubation with 20 µM GDC-0980 (Figure 4B), where 46.47% of apoptotic cells and only 52.50% of viable cells were observed. The pro-apoptotic activity of GDC-0980 was slightly lower in U-118-MG cell line. The weakest induction of apoptosis was shown by the U-118-MG cell line after 48 h of incubation with GDC-0980 (Figure 4D), where the percentage of apoptotic cells was only 6.2% with 10 µM of GDC-0980 and only 7.7% apoptotic cells with 20 µM GDC-0980. Based on the obtained results, we concluded that the apoptotic activity of GDC-0980 depends on the sensitivity of GBM cell lines and may depend on the method of induction of programmed cell death (Figure 4A–D).

### 2.3. GDC-0980 Treatment Inhibits the PI3K/mTOR Pathway

The effect of GDC-0980 on the activity of PI3K/mTOR signalling pathway was then investigated in A-172 and U-118-MG glioblastoma cell lines. We confirmed inhibition of the PI3K/mTOR pathway following GDC-0980 treatment by examining the S6K, AKT, and mTOR phosphorylation by Western blot analysis. It was found that GDC-0980 treatment evoked a significant reduction in the levels of phosphorylated S6K^Thr421/Ser424^, AKT^Ser473^, and mTOR^Ser2448^ in both A-172 and U118-MG cell line (Figure 5). Taking these results together, we confirmed that GDC-0980 blocked the phosphorylation of S6K^Thr421/Ser424^, AKT^Ser473^, and mTOR^Ser2448^ and prevented their activation, which resulted in inhibition of the PI3K/mTOR signalling pathway in both investigated GBM cell lines.

### 2.4. The Effect of GDC-0980 Treatment on the Expression of Important Markers of ER Stress

Endoplasmic reticulum (ER) stress caused by misfolded proteins or cytotoxic drugs can kill cells. Inhibition of PI3K/mTOR signalling pathway is also very important source of ER stress. Therefore, we tested whether GDC-0980 induced ER stress by PI3K/mTOR inhibition. To investigate this phenomenon, the expression of the important ER stress markers PERK, EIF2α, and phospho-EIF2α^Ser51^ was analysed in A-172 and U-118-MG cell lines treated with 10 and 20 µM GDC-0980 for 48 h by Western blot method. We found that GDC-0980 treatment decreased the expression of PERK and reduced the ratio between the means of densitometric values of P-EIF2α/total EIF2α in both cell lines (Figure 6 and Appendix A).

### 2.5. The Effect of GDC-0980 Treatment on the Expression of Important Markers of Apoptosis

Figure 7 shows that treatment with GDC-0980 resulted in enhanced accumulation of the proapoptotic transcription factor CHOP (C/EBP-homologous protein), although, to a different extent, in a cell-dependent manner. In A-172 cells, a more than 20-fold increase in CHOP expression was observed compared to control cells. Interestingly, in U-118-MG, this expression only increased three-fold. The enhanced expression of CHOP confirms the presence of apoptosis in cells subjected to GDC-0980 treatment (Figure 7). In addition, the treatment with GDC-0980 resulted in increased PARP (polyADP-ribose protein) accumulation, although to a varying extent in a cell-dependent manner (Figure 7).

To further explore the mechanism of cell death in investigated cells, we assayed the cells for evidence of caspase-dependent apoptosis. The activation of apoptosis leads to proteolytic cleavage and thus activation of caspases, a family of cysteine proteases that act as common death effector molecules. This leads to caspase-8 activation, which can then directly cleave downstream effector caspases such as caspase-3. GDC-0980 exposure in A-172 cell line was accompanied by activation of caspase-3, caspase-7, and casapase-8, while in U-118-MG, we only observed the cleavage of caspase-3 and caspase 8 (Figure 7). Alternatively, caspase-8 can promote outer mitochondrial membrane permeabilisation by cleaving Bid, a BH3-only protein that translocates to mitochondria upon cleavage and causes cytochrome c release. In our study, we confirmed that BID expression was decreased in both cell lines and associated with increased expression of proapoptotic protein BAX (Figure 8). Additionally, treatment with GDC-0980 resulted in enhanced accumulation of antiapoptotic BCL-2 proteins, although, to a different extent, in a cell-dependent manner. In U-118-MG cell line the expression of antiapoptotic proteins BCL-2 was higher than in A-172 (Figure 7). Our results clearly suggest that GDC-0980 evokes death receptor (extrinsic) pathway of apoptosis, but we did not observe enhanced expression of death receptor 5 (DR5) in both cell lines (Figure 7).

### 2.6. GDC-0980 Evokes Autophagosome Formation in GBM Cells

Figure 8 shows the light chain expression of autophagy marker 3 (LC3A/B-I and LC3A/B-II) in A-172 and U-118-MG cells incubated with 10 and 20 µM GDC-0980 for 48 h. A-172 cells incubated for 48 h with 10 and 20 µM GDC-0980 express the LC3A/B-I form and demonstrate only weak expression of LC3A/B-II. Furthermore, we observed that expression of the membrane-bound, autophagic vacuole form, LC3A/B-II was increased in U-118-MG cells incubated with 10 and 20 µM GDC-0980 for 48 h compared to control cells. Then, we performed the densitometric analysis and calculated the ratio LC3A/B-II/LC3A/B-I (Figure 8). We observed a time- and dose-dependent increase in the LC3A/B-II/LC3A/B-I ratio in both cell lines. Moreover, in line A-172, it was only a slight increase in the ratio LC3A/B-II/LC3A/B-I, while in line U-118-MG, a more than 2.5-fold increase in this ratio was observed compared to the control cells. Increased LC3A/B-II/LC3A/B-I ratio suggests autophagosome formation in GDC-0980 treated cells.

## 3. Discussion

GBM is considered to be among the most lethal cancer, characterised by a rapid growth and extensive infiltration to neighbouring brain tissues [1]. The standard treatment options available to patients are usually ineffective. Therefore, additional investigations are urgently needed to identify therapeutic targets in GBM and design new therapeutic strategies for the treatment of GBM [1,14].

The antitumor activity of compounds inhibiting the PI3K/AKT/mTOR pathway has been confirmed in a number of reports [6,14,15,16]. Most of the clinical trials conducted so far concern the use of mTOR kinase inhibitors [17,18]. Additionally, cancer cells still maintain mTOR activation even although the actions of PI3K and AKT are suppressed. These crosstalk and feedbacks between mTOR andPI3K greatly reduce the therapeutic effects of mTOR or PI3K inhibitors. Therefore, dual PI3K/mTOR inhibitors such as NVP-BEZ235, XL765, GDC-0084, and PQR309 are produced and currently tested in clinical trials [1].

Moreover, one of the most important elements that evokes dysfunction of the of the PI3K/AKT/mTOR signalling pathway is the suppressor protein PTEN (phosphatase and tensin homolog deleted on chromosome 10); it has often been cited as the second most frequently mutated gene in cancer and even as “a new guardian of the genome” [19]. PTEN negatively regulates the PI3K/AKT pathway. Loss of PTEN activity was demonstrated in many cases of GBM [1,17], and loss of PTEN activity leads to permanent activation of the PI3K/AKT pathway. The present studies were carried out on two established glioblastoma cell lines: A-172 and U-118-MG. Both of them contain genomic mutations in the negative regulator of PI3K, PTEN (Sanger COSMIC database). There is also an additional TP53 mutation in the U-118-M cell line [20]. These various degrees of genetic complexity make available new therapies for GBM by targeting these pathways with selective inhibitors [1].

GDC-0980 is dual PI3K/mTOR inhibitor that targets both PI3K and mTOR in the PI3K/AKT/mTOR signalling pathway [7]. GDC-0980 have entered phase I/II clinical trials for advanced solid tumours. However, no clinical study on these inhibitors has been carried out in patients with GBM [1].

In our study, we demonstrated that GDC-0980 exerts cytotoxic effect on both A-172 and U-118-MG glioblastoma cells. The MTT test indicated that GDC-0980 reduced these cells viability in a concentration-dependent manner. Different from previous described mTOR inhibitors, GDC-0980 not only suppressed the GBM cell growth but also demonstrated an obvious killing effect on GBM cells.

In our research, we confirmed that GDC-0980 inhibited PI3K-AKT-mTOR signalling in tested human GBM cell lines. The results show that GDC-0980 treatment evoked a significant reduction in expression of phosphorylated: S6K^Thr421/Ser424^, AKT^Ser473^ and mTOR^Ser2448^ in both cell lines.

Inhibition of PI3K/mTOR signalling pathway in cancers is also very important source of ER (endoplasmic reticulum) stress and subsequent UPR (unfolded protein response) activation, thanks to which cancer cells survive under severe physiological and cellular conditions [4]. PERK (RNA-dependent protein kinase-like ER kinase), a kinase localised in the ER membrane, phosphorylates the α subunit of the eukaryotic initiation factor of translation (EIF2α) at the Ser51 site, in turn leading to translation attenuation and then preventing further accumulation of unfolded proteins [21,22]. We found that GDC-0980 treatment resulted in decreased expression of PERK and reduced the ratio of P-EIF2α/total EIF2α, although, to a variable extent, in a cell-dependent manner. We suggest that GDC-0980, by exerting an inhibitory effect on PERK expression, may thus block its inhibitory effect on protein synthesis, leading to intensification of translation, and this may result in an increase in apoptosis. Therefore, the development of clinically relevant ER stress inducers and PERK inhibitors holds promise as therapeutic strategies in GBM.

Apoptosis is a very complex mechanism that is tightly regulated at many levels in the cell. Such control protects the cell from undesirable diversion into apoptosis, but on the other hand, makes the damaged cell insensitive to factors that normally induce apoptosis. The loss of the ability to induce programmed cell death is one of the key events enabling the formation and growth of cancer cells. In this study, we first discovered that dual inhibitor GDC-0980 could block GBM cell survival by inducing apoptosis in tested human GBM cell lines in a time- and dose-dependent manner. Treatment with GDC-0980 resulted in enhanced accumulation of CHOP and PARP, although, to a different extent, in a cell-dependent manner. These results are in line with previous reports by Penaranda-Fajardo et al., demonstrating that PERK inhibition was associated with a reduction in PARP cleavage [23]. We suggest that inhibition of PERK expression in our experimental conditions, evoked by GDC-0980 in GBM cell lines, is accompanied by inhibition of PARP cleavage. On the other hand, increased expression of CHOP is induced by certain cellular stresses, including cancer. CHOP leads to the activation of GADD34, which, in turn, dephosphorylates phospho-Ser51 of EIF2α, thereby stimulating protein synthesis and leading to an increase in apoptosis.

Moreover, we have shown an elevated activity of caspase-3, caspase-7, and caspase-8 in investigated cells after treatment with GDC-0980, confirming apoptosis to be a major phenomenon triggered by stimulation with this inhibitor. GDC-0980 exposure was accompanied by caspase-3, caspase-7, and casapase-8 activation in A-172 cell line, but in U-118-MG, we identified only cleavage of caspase-3 and caspase 8. We observed that this was associated with increased expression of proapoptotic protein BAX, but BID expression was decreased in both cell lines. Interestingly, GDC-0980 treatment resulted in enhanced accumulation of anti-apoptotic proteins BCL-2, although, to a variable extent, in a cell-dependent manner.

It has already been demonstrated that dual PI3K/mTOR inhibitors induced apoptosis through the activation of death receptors including FAS, TNFαR, DR3, DR4, and DR5 by their respective ligands [24]. He et al. demonstrated that death receptor 5 (DR5) plays a critical role in ER stress-induced apoptosis [24]; we therefore verified its regulation in response to GDC-0980 stimulation. Interestingly, in our research, in both investigated cell lines subjected to GDC-0980 treatment, we did not observe enhanced expression of DR5.

Likewise, Cai et al. reported that inhibition of the activities of ER stress markers slowed tumour cell growth and activated expression of CHOP and caspase 7 in temozolomide (TMZ) treated cells [4,24,25]. We found that inhibition of PI3K/mTOR by GDC-0980 induced apoptosis in GBM cells, which was independent on the ER stress-mediated DR5 activation. Although apoptosis seems to be predominant mechanism activated after exposition to GDC-0980, inherent heterogeneity of GBM cells necessitates further examination of possible pathways activated after GDC-0980 stimulation.

Moreover, in both investigated glioma cell lines, in addition to apoptosis, we observed hallmarks of autophagy, a catabolic pathway that degrades aged or damaged organelles. During the formation of the autophagosome, cytosolic microtubule-associated protein 1 light chain 3-I (LC3I) is cleaved and conjugated with phosphatidylethanolamine (PE), leading to formation of LC3II, an autophagic vacuole-associated form. Thus, an increase in the amount of the LC3II protein and an increase in the LC3II/LC3I ratio have been considered hallmarks of autophagy. We noticed a time- and dose-dependent increase in LC3A/B-II/LC3A/B-I ratio in both cell lines. The increased LC3A/B-II/LC3A/B-I ratio suggests the occurrence hallmarks of autophagy in both cell lines subjected to GDC-0980 treatment.

In cancer cells, mTOR serves as the major negative regulator of autophagy. This protein is a 289 kDa serine/threonine protein kinase, localised in two structurally and functionally distinct multiprotein complexes in mammalian cells, the mammalian target of rapamycin complex 1 (mTORC1) and the mammalian target of rapamycin complex 2 (mTORC2). mTORC1 contributes to tumour growth by inhibiting autophagy [26].

mTORC1 suppresses autophagy directly by inhibiting the kinase complex unc-51-like kinase 1/mammalian autophagy-related gene 13/focal adhesion kinase family-interacting protein of 200 kDa (ULK1/Atg13/FIP200), which is a key component required for the autophagy induction, and indirectly by modulating the expression of death-associated protein 1 (DAP1), a novel substrate of mTORC1 that negatively regulates autophagy. Inhibition of mTORC1 induces autophagy [2,26]. We demonstrated that investigated GBM cells treated with GDC-0980 showed a reduction in serine 2448 mTOR phosphorylation along with decreased phosphorylated S6K^Thr421/Ser424^, an effector downstream of the mTOR pathway that activates the synthesis of protein and growth [21], indicating that GDC-0980 inhibits mTORC1, which consequently leads to induction of autophagy.

In this study, we first discovered that dual PI3K/mTOR blockade by GDC-0980 markedly supressed survival of tested human GBM cell lines. We suggest that GDC-0980, by exerting an inhibitory effect on PERK expression, may thus block its inhibitory effect on protein synthesis, leading to its intensification, and this may result in an increase in apoptosis. On the other hand, CHOP leads to the activation of GADD34, which, in turn, dephosphorylates phospho-Ser^51^ of EIF2α thereby stimulating protein synthesis and it functions to increase in apoptosis. We found that inhibition of PI3K/mTOR by GDC-0980 induced apoptosis in GBM cells, which was independent on the ER stress-mediated DR5 activation. However, promising results obtained in vitro should be tested in vivo to fully elucidate the anticancer effectiveness of GDC-0980. In summary, GDC-0980 is a potent small molecule inhibitor of class I PI3K and mTOR kinase with promise for clinical trials in glioblastoma multiforme as either a single agent or in combination with antimitotic agents.

## 4. Materials and Methods

### 4.1. Reagents

Dulbecco’s modified Eagle’s medium (DMEM), containing glucose at 4.5 mg/mL with GlutaMax^TM^, trypsin-EDTA, penicillin, streptomycin, and fetal bovine serum Gold (FBS Gold), were provided by Gibco (San Diego, CA, USA). RIPA buffer and Protease/Phosphatase Inhibitor Cocktail were provided by Cell Signalling Technology (Boston, MA, USA) and BCA Protein Assay Kit by Thermo Scientific (Rockford, IL, USA). Apitolisib (GDC-0980, PI3K/mTOR inhibitor) was a product of MedChem Express, Annexin V Apoptosis Detection Kit I by BD Pharmingen^TM^ (San Diego, CA, USA) and Immuno-Blot PVDF Membranes for Protein Blotting by Bio-Rad (Hercules, CA, USA). Monoclonal (rabbit) anti-human CHOP antibody, polyclonal (rabbit) anti-human PARP antibody, polyclonal (rabbit) anti-human cleaved caspase-3 antibody, monoclonal (mouse) anti-human cleaved caspase-8 antibody, monoclonal (rabbit) anti-human cleaved caspase-9 antibody, monoclonal (rabbit) anti-human DR-5 antibody, polyclonal (rabbit) anti-human BAX antibody, monoclonal (rabbit) anti-human BCL-2 antibody, monoclonal (rabbit) anti-human PERK antibody, polyclonal (rabbit) anti-human EIF2α antibody, monoclonal (rabbit) anti-human Phospho-EIF2α^Ser51^ antibody, polyclonal (rabbit) anti-human Phospho-p70S6K^Thr421/Ser424^ antibody, monoclonal (rabbit) anti-human Phospho-AKT^Ser473^ antibody, monoclonal (rabbit) anti-human Phospho-mTOR^Ser2448^ antibody, polyclonal (rabbit) anti-human LC3A/B antibody, polyclonal (rabbit) anti-human β-tubulin antibody, and alkaline phosphatase-labeled anti-rabbit immunoglobulin G were provided by Cell Signaling Technology (Boston, MA, USA).

### 4.2. Cell Cultures and Exposure to GDC-0980

The A-172 and U-118-MG cell lines were provided by American Type Culture Collection (ATCC) (Rockville, MD, USA). These cells were characterised by varying degrees of genetic complexity. A-172 culture is not tumorigenic and contains genomic mutations of two genes, CDKN2A and PTEN. U-118-MG culture is tumorigenic and contains genomic mutations of the following genes: CDKN2A, PTEN, and TP53 (Sanger COSMIC database). 

The cells were cultured in DMEM, supplemented with heat-inactivated, 10% (FBS Gold) streptomycin (100 μg/mL) and penicillin (100 U/mL), as we described previously [27]. Next, 2.0 × 10^5^ cells were seeded in 2 mL of DMEM in six-well plates. After 24 h incubation, DMEM was removed and replaced with DMEM containing GDC-0980 at concentrations ranging from 5 to 50 µM. The A-172 and U-118-MG cells not treated with GDC-0980 served as the negative controls. Next, the cells were incubated for 24 and 48 h and retained for further analyses.

### 4.3. Cell Viability/MTT Assay

Cell viability was measured according to the manner of Carmichael et al. using 3-(4,5-dimethylthiazol-2-yl)-2,5-diphenyltetrazolium bromide (MTT) [28]. The A-172 and U-118MG cells, at a density of 2.0 × 10^5^ per well, were seeded in 6-well plates. After 24 h, DMEM was removed and replaced with DMEM containing GDC-0980 at concentrations ranging from 50 to 50,000 nM. The untreated A-172 and U-118-MG cells served as the negative controls. Then, both cell lines were incubated with GDC-0980 for 24 and 48 h, as we described previously [27]. The viability of A-172 and U-118-MG cells cultured with GDC-0980 was calculated as the percentage of the untreated cells. All the experiments were done in duplicates in at least three cultures.

### 4.4. Proliferation Test/Crystal Violet Assay

The A-172 and U-118-MG cells were seeded (0.5 × 10^5^ per well) in 1 mL of medium in 24-well culture plate and allowed to adhere for 24 h at 37 °C. Next, the cells were treated with various, 10 and 20 μM, concentrations of GDC-0980. Cell lines were incubated for 24 and 48 h. After, they were washed with cold PBS and then fixed in 1% formaldehyde in PBS for 10 min at room temperature. After fixation, the cells were permeabilised in 1% methanol for 5 min, and the cells were stained with a mixture of 0.05% crystal violet. Next, the intracellular crystal violet product was dissolved in methanol, and absorbance was measured at the wavelength of 590 nm in a microplate reader (Tecan, Männedorf, Switzerland). The proliferation of GDC-0980-treated A-172 and U-118-MG cells was calculated as a percentage of control untreated cells. All the experiments were performed in duplicate in at least three cultures.

### 4.5. Detection of Apoptosis and Necrosis

Apoptosis and necrosis of A-172 and U-118-MG cell lines were evaluated by flow cytometry on FACSCanto II cytometer (BD, San Diego, CA, USA) as we described previously [27]. 

### 4.6. Western Blot Analysis

The A-172 and U-118-MG cells were washed with cold PBS and solubilised in 100 μL per well of RIPA buffer enriched with a protease-/phosphatase-inhibitor cocktail (100×). The lysates from each well were centrifuged at 10,000× *g*, at 4 °C, for 10 min. Samples of lysates containing 20 μg of protein were subjected to SDS-PAGE, as described by [29]. The 10% or 12% polyacrylamide gel and constant current (25 mA) were used.

The proteins were transferred to PVDF membranes for protein blotting and subsequently pre-treated with Tris-buffered saline (TBS) at room temperature for 1 h. Each strip was treated with blocking buffer (5% nonfat dry milk in TBS containing 0.05% Tween 20 (TBS-T), pH 7.6 for 1 h. The membranes were washed for 15 min and 4 times for 5 min in TBST, and next were probed with the following antibodies at the indicated concentrations: polyclonal (rabbit) anti-human PARP antibody (1:1000); monoclonal (rabbit) anti-human CHOP antibody (1:1000); polyclonal (rabbit) anti-human cleaved caspase-3 antibody (1:1000); monoclonal (mouse) anti-human cleaved caspase-8 antibody (1:1000); monoclonal (rabbit) anti-human cleaved caspase-9 antibody (1:1000); monoclonal (rabbit) anti-human DR-5 antibody (1:1000); polyclonal (rabbit) anti-human BAX antibody (1:1000); monoclonal (rabbit) anti-human BCL-2 antibody (1:1000); monoclonal (rabbit) anti-human PERK antibody (1:1000); polyclonal (rabbit) anti-human EIF2α antibody (1:1000); monoclonal (rabbit) anti-human Phospho-EIF2α^Ser51^ antibody (1:1000); polyclonal (rabbit) anti-human Phospho-p70S6K^Thr421/Ser424^ antibody (1:1000); monoclonal (rabbit) anti-human Phospho-AKT^Ser473^ antibody (1:2000); monoclonal (rabbit) anti-human Phospho-mTOR^Ser2448^ antibody (1:1000); polyclonal (rabbit) anti-human LC3A/B antibody (1:1000); and polyclonal (rabbit) anti-human β-tubulin antibody (1:1000) in TBS with 5% non-fat dry milk, at 4 °C for 16 h.

After washing with TBS-Tween 20 for 15 min and 2 times for 5 min, each strip was incubated with horseradish peroxidase (HRP) conjugated with anti-human secondary antibody against rabbit or mouse IgG (whole molecule) at 1:1000 dilution, in TBS, with slow shaking for 2 h. The membranes were washed for 15 min and 4 times for 5 min with TBS-T and exposed to ECL reagent in dark for chemiluminescence detection.

### 4.7. Chemiluminescence Detection

For chemiluminescence detection, WB membranes were incubated in the dark with ECL Reagent (Cell Signaling), which is a luminol-based enhanced chemiluminescence substrate for peroxidase. The luminescent signal was recorded and quantified with the GeneGnome (Syngene, Frederick, MD, USA). The luminescent signals were detected and transmitted to the GeneTools software (Syngene, Frederick, MD, USA) for analysis and documentation.

### 4.8. Protein Assay

Protein concentration in cell lysates was measured by the method of Smith et al. using BCA Protein Assay Kit [30]. Bovine serum albumin was used as a standard. 

### 4.9. Statistical Analysis

STATISTICA version 13.3 program (StatSoft, Inc., Tulsa, OK, USA) and GraphPad Prism software version 9 (GraphPad Software, San Diego, CA, USA) were used for statistical analysis. Data were presented as means, standard deviations (SD), and percentages (%). Data were statistically analysed by one-way analysis of variance (ANOVA). Duncan’s test was used as a post hoc test for the comparison of significance between groups. A *p* value < 0.05 was considered statistically significant.

## Figures and Tables

**Figure 1 ijms-22-11511-f001:**
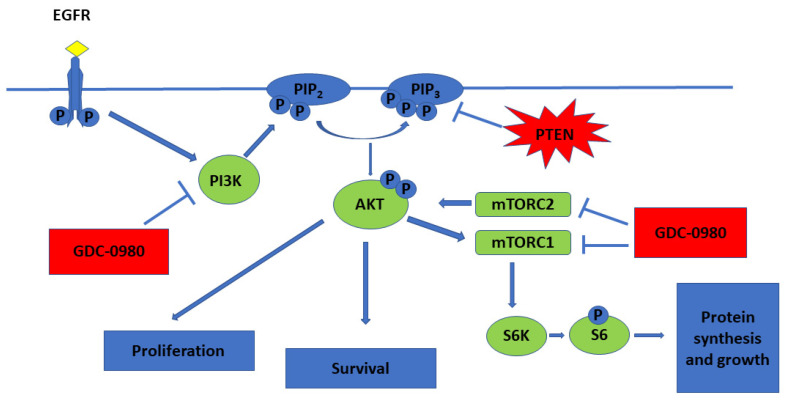
PI3K/mTOR pathway and mechanistic effects of GDC-0980 treatment in tumour cell lines (
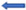
 activation; 
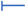
 inhibition).

**Figure 2 ijms-22-11511-f002:**
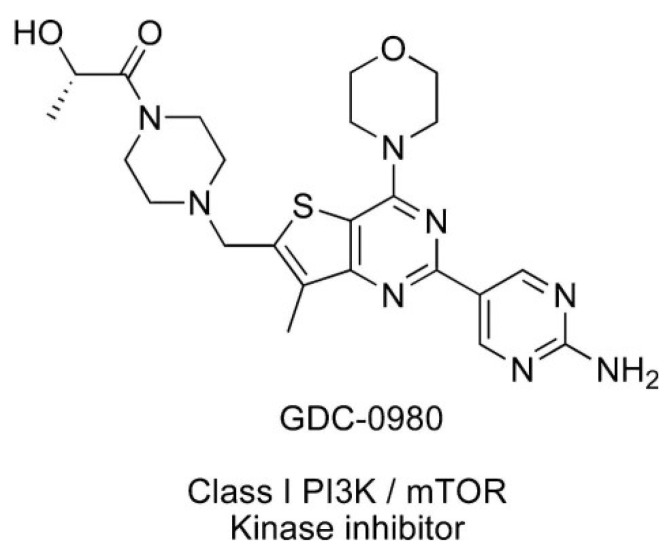
Structure of Apitolisib (GDC-0980), morpholino thienopyrimidine inhibitor of class I PI3K and mTOR kinase in the PI3K/mTOR signalling pathway [12]. (http://creativecommons.org/publicdomain/zero/1.0/).

**Figure 3 ijms-22-11511-f003:**
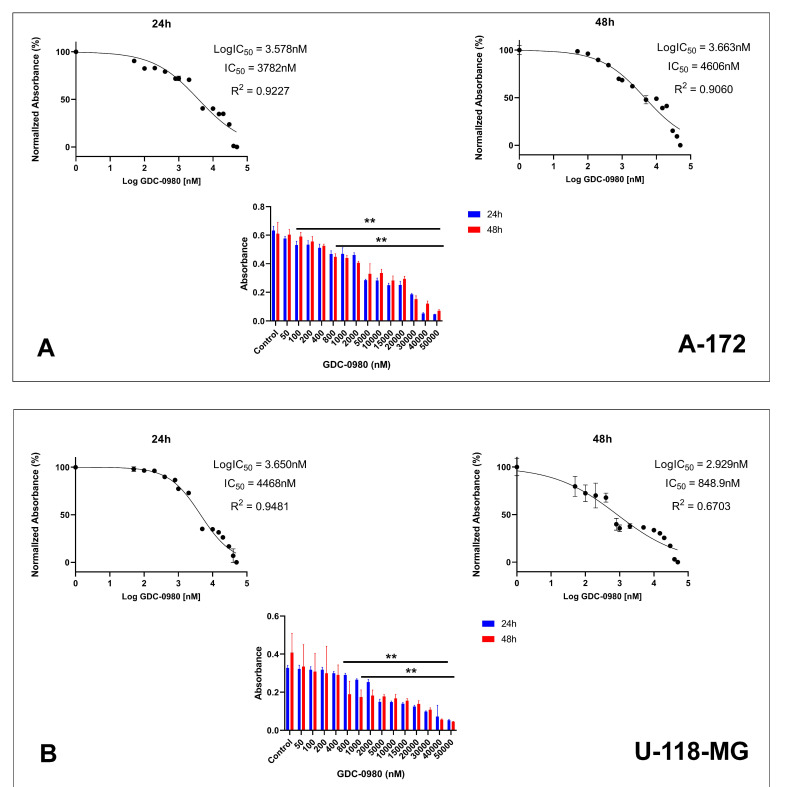
The viability of A-172 (**A**) and U-118-MG (**B**) glioblastoma cell lines treated with different concentrations of GDC-0980 ranging from 50 to 50,000 nM (**A**,**B**) and the crystal violet assay analysis for the proliferation of A-172 (**C**) and U-118-MG (**D**) cells treated with 10 and 20 µM concentrations of GDC-0980 for 24 and 48 h. Mean values from three independent experiments ± SD are presented. Note: significant alterations are marked with asterisks: * *p* < 0.05; ** *p* < 0.01. Statistical significance was considered if *p* < 0.05.

**Figure 4 ijms-22-11511-f004:**
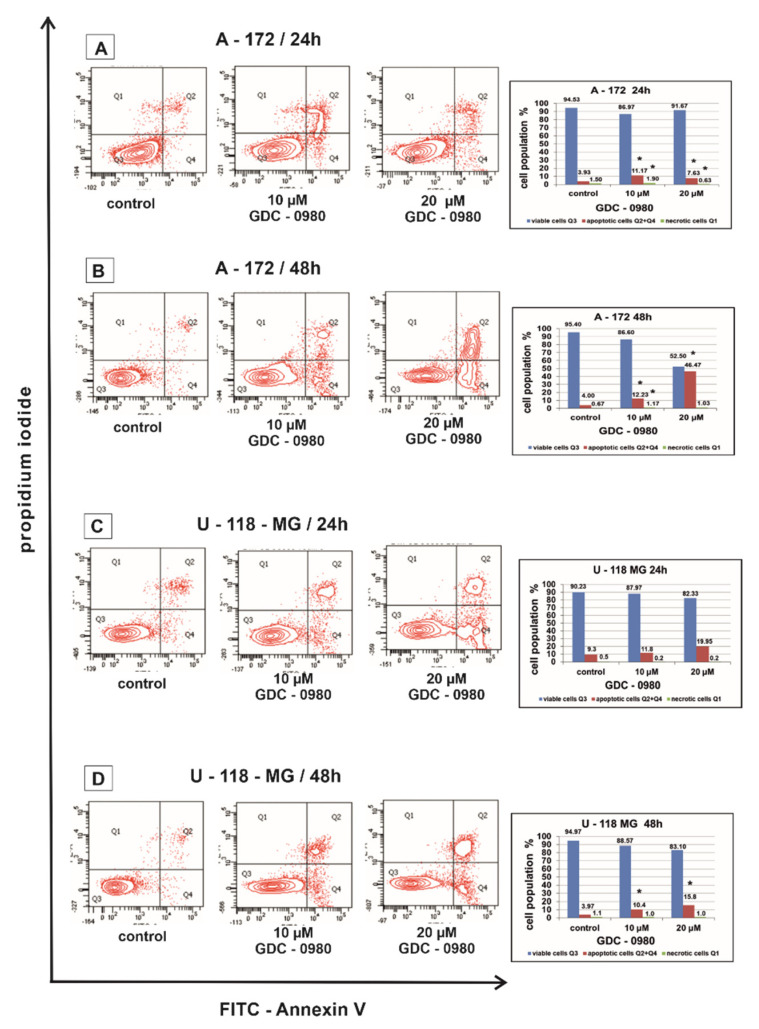
The effect of GDC-0980 inhibitor on apoptosis and necrosis of A-172 (**A**,**B**) and U-118-MG (**C**,**D**) cells evaluated by Annexin V assay. The cells were incubated for 24 and 48 h in Dulbecco’s modified Eagle medium (DMEM) with 10 and 20 µM of GDC-0980. The cells were double-stained with Fluorescein Isothiocyanate (FITC)-Annexin V and propidium iodide (PI). Representative flow cytometry (FACS) analysis via Annexin V-FITC/PI staining for 24 h and 48 h is presented. The bar graphs present the percentage of apoptotic cells as a sum of Q2 and Q4 quadrants and necrotic cells as a Q1 quadrant of the analysed cell population. Mean values of the percentage of apoptotic and necrotic cells from three independent experiments ± SD are presented. Note: significant alterations are expressed relative to controls and marked with asterisks: * *p* < 0.05. Statistical significance was considered if *p* < 0.05.

**Figure 5 ijms-22-11511-f005:**
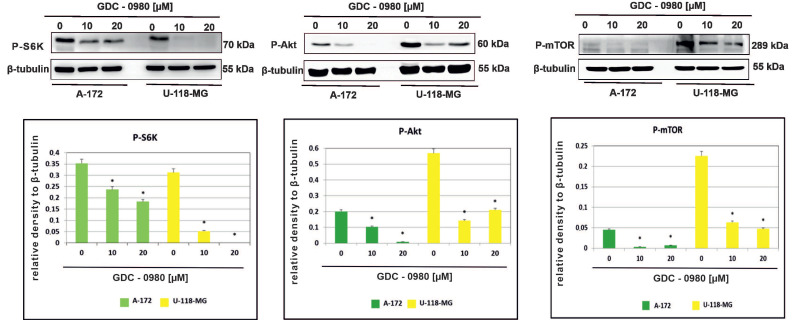
GDC-0980 inhibits PI3K pathway signalling in A-172 and U-118 MG glioblastoma cell lines. Western blot data show the relative densities of P-S6K^Thr421/Ser424^, P-AKT^Ser473^, and P-mTOR^Ser2448^ to β-tubulin after 48 h of GDC-0980 treatment. β-tubulin (50 kDa) was used as a sample of loading control. Samples containing 20 μg of protein were submitted to electrophoresis and immunoblotting. The representative bands of the P-S6K^Thr421/Ser424^, P-AKT^Ser473^, and P-mTOR^Ser2448^ to β-tubulin are illustrated. Data are presented as the means ± SEM. * *p* < 0.05.

**Figure 6 ijms-22-11511-f006:**
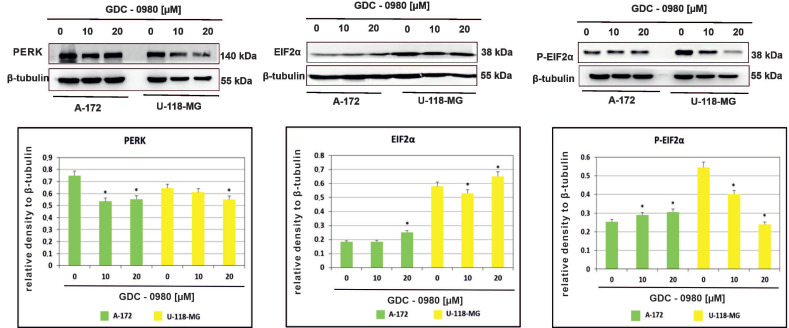
Western blot analysis demonstrates the effect of 48 h GDC-0980 treatment on the expression of PERK, EiF2α, and P-EiF2α^Ser51^ in A-172 and U-118 MG glioblastoma cells, using β-tubulin as a protein-loading control. Samples containing 20 μg of protein were submitted to electrophoresis and immunoblotting. The representative bands of the PERK, EiF2α, and P-EiF2α^Ser51^ to β-tubulin are illustrated. Data are presented as the means ± SEM. * *p* < 0.05.

**Figure 7 ijms-22-11511-f007:**
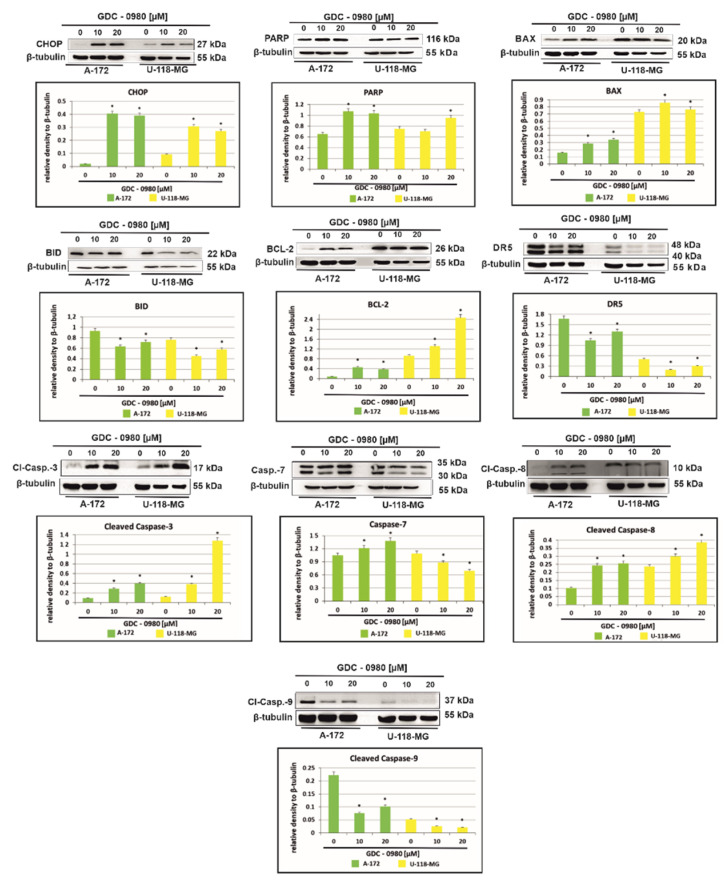
Western blot analysis demonstrates the effect of 48 h GDC-0980 treatment on the expression of CHOP, PARP, BAX, BID, BCL-2, DR5, cleaved caspase-3, caspase-7, cleaved caspase-8, and cleaved caspase-9 in A-172 and U-118 MG glioblastoma cells, using β-tubulin as a protein loading control. The representative bands of the CHOP, PARP, BAX, BID, BCL-2, DR5, cleaved caspase-3, caspase-7, cleaved caspase-8, and cleaved caspase-9 to β-tubulin are illustrated. Samples containing 20 μg of protein were submitted to electrophoresis and immunoblotting. Data are presented as the means ± SEM. * *p* < 0.05.

**Figure 8 ijms-22-11511-f008:**
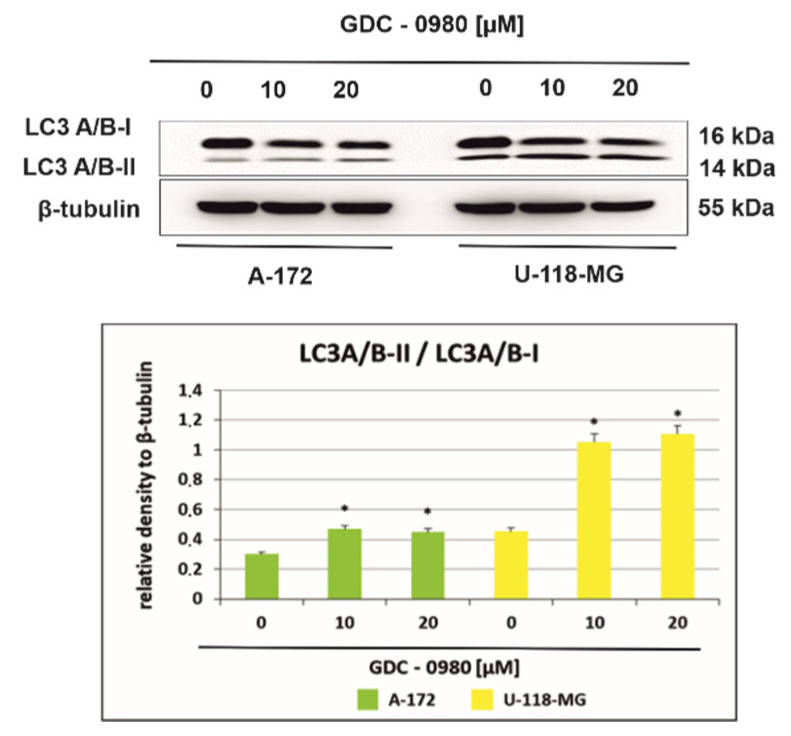
Western blot analysis demonstrates the effect of 48 h GDC-0980 treatment on the expression of autophagy marker Light Chain 3 (LC3A/B-I and LC3A/B-II) expression in A-172 and U-118 MG glioblastoma cells, using β-tubulin as a protein-loading control. The representative bands of the LC3A/B-I and LC3A/B-II to β-tubulin are illustrated. Samples containing 20 μg of protein were submitted to electrophoresis and immunoblotting. Data are presented as the means ± SEM. * *p* < 0.05.

## Data Availability

Not applicable.

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
