# Peer review of "Novel Dual PI3K/mTOR Inhibitor, Apitolisib (GDC-0980), Inhibits Growth and Induces Apoptosis in Human Glioblastoma Cells"

_ijms, 2021, doi:10.3390/ijms222111511_

Round 1
Reviewer 1 Report
This paper suggests the mechanism of GDC-0980 action on human glioblastoma cells survival in vitro.
The main concern is about the relevance of the choice of the studied drug. One of the problems in brain cancer treatment is the low drugs penetrance into the brain due to blood-brain barrier. The in vitro studies may demonstrate the great suppression of GBM cells survival upon GDC-0980 treatment however if it is not able to cross blood brain barrier the significance of the data obtained would be low. According to the literature, GDC-0980 is subject to active efflux at the BBB so it would be hard to reach high concentration of the drug in the brain in vivo, and thus its efficacy in GBM treatment is questionable (Becker et al. Neuro Oncol. 2015;17(9):1210-1219.; Tarantelli C. et al. Int J Mol Sci. 2020;21(3):1060.). Authors should clarify why the GDC-0980 was chosen for the study.
The other concerns regard:
- In the Introduction authors state “the treatment is palliative only” however this statement is false. The conventional glioblastoma treatment include surgery, radiotherapy and chemotherapy sometimes in combination with novel treatment strategies such as immunotherapy or target therapy. Authors should research this matter and add a state-of-art citation.
- The last 2 paragraphs of the introduction include the redundant repetition of the information (lines 63-65 and 77-78; lines 69,70 and 78,79).
- Authors state the aim of the study twice and the aim descriptions do not match (lines 75-77 and 83-85)
- The paragraphs 2.2, 2.3, 2.4 and 2.5 contain redundant information and should be shortened.
- The paragraph 3.1 should be shortened, as the drug demonstrates the same effect on the both cell lines. It is not clear why authors describe 2000nM concentration as “nM concentration” and 5uM as “uM concentration”.
- The description of the figure 3 lacks information: it is unclear which graph corresponds to which experiment. The histogram plot in Fig 3 is confusing as it shows significant differences between different drug concentrations but description says everything was compared to control.
- Figure 4 is not informative and can be transformed into a panel in figure 3 or transferred to supplementary material with no loss for the paper.
- The reason to study of the ER stress is not well explained. Some information on this matter should be added either to introduction section or to the beginning of the 3.4 paragraph.
- The second paragraph in 3.5 is difficult to understand and should be re-written.
- The vast part of the Discussion section repeats the information from the introduction (lines 360-388).
- Discussion lines 400-409. Authors showed that the drug used in the study inhibit the PI3K-AKT-mTOR signaling. The cited literature about AKT, mTOR, and 404 S6K phosphorylation in high- and low-grade glioma does not seem to be relevant to this data.
Minor points:
Panels in figures can be reorganized to decrease the height of the figures and thus place them closer to the corresponding text. For example, in Fig. 8 panels can be put four in a row instead of just two.
Table 1 can be transferred to supplementary materials
Line 280 – abbreviation ER is used without definition (the definition appears only in the discussion section)
Reviewer 2 Report
The manuscript by Omeljaniuk and colleagues reports that a dual inhibition PI3K/mTOR signaling induces cytotoxicity and apoptosis in two glioma cell lines. The study is interesting. Only a minor point in the paragraph 2.6. Western blot analysis need to be added the sources of antibodies.
Round 2
Reviewer 1 Report
Authors have fully answered the questions from my previous review. The last version of the paper meets the high standards of the journal and can be accepted for publication.